# Comparison of Postural Stability and Regulation among Male Athletes from Different Sports

Andreas Lauenroth [1], Stephan Schulze [1], Lars Reinhardt [1], Kevin G. Laudner [2], Karl-Stefan Delank [1] and René Schwesig [1,*]

1  Department of Orthopaedic and Trauma Surgery, Martin-Luther-University Halle-Wittenberg, Ernst-Grube-Str. 40, 06120 Halle (Saale), Germany; lauenroth@univations.de (A.L.); stephan.schulze@uk-halle.de (S.S.); lars.reinhardt@uk-halle.de (L.R.); stefan.delank@uk-halle.de (K.-S.D.)
2  Department of Health Sciences, University of Colorado, Colorado Springs, CO 80918, USA; klaudner@uccs.edu
*  Correspondence: rene.schwesig@uk-halle.de; Tel.: +49-345-557-4897; Fax: +49-345-557-4899

**Featured Application: In professional sports, the degree of postural control depends on the sport's specific demands necessary to be successful. The present study compared different male sports athletes (ice hockey, soccer, diving) using posturography under four conditions. Significant differences were observed between soccer players and divers regarding postural stability and medio-lateral weight distribution, especially under reduced somatosensory conditions. These findings provide coaches and athletes with more evidence to identify deficits in posture stability and regulation as a potential risk of injury and decreased performance.**

**Abstract:** The purpose of this study was to assess the postural control of 50 male athletes (age: 24.9 ± 4.55 years) who participate in different elite-level competitive sports. Athletes from two team sports from the third German league (ice hockey: $n = 16$; soccer: $n = 23$), and one individual sports (diving: $n = 11$) were included. These athletes were investigated using posturography under different conditions (e.g., stable surface, unstable surface; eyes open, eyes closed) to determine postural stability and regulation. Most of the performance maxima were found among the divers (6), followed by ice hockey (5) and soccer (4). The biggest effect of sport was found in the stability indicator, where the subject was standing on a stable surface and their eyes were closed (NC; $p = 0.001$, $\eta_p^2 = 0.273$). This significance was observed between the soccer (17.3 ± 5.66) and diving (24.9 ± 6.98) subjects. The stability indicator had the largest significant effect (50%, 2/4). These results provide coaches and athletes insight into the postural stability and regulation of male athletes in sports with different demands on postural control. Especially for soccer players, it may be beneficial to address muscular imbalances to reduce the risk of lower extremity injuries.

**Keywords:** performance diagnostic; team sports; postural regulation; posturography





## 1. Introduction

Postural control involves a complex integration of the entire sensorimotor system to maintain an individual's center of mass over their base of support [1]. Furthermore, this system must constantly adapt to changing environments caused by uneven surfaces, decreases in sensory input, as well as acceleration and deceleration forces (just to name a few) [2]. Not only is this control critical in activities of daily living to prevent falls, but this system can be taxed even more during competitive athletics when the kinematics and kinetics required by a particular sport are increased substantially. As such, deficiencies in an athlete's postural control can result in decreased performance and potential injury [3,4].

Postural regulation is critical for athletes regardless of sport. Whether it is athletes who are required to change directions quickly, such as soccer, basketball, handball, tennis, etc. [3,5–7], athletes who produce highly-skilled technical movements, which are often

scored by a judge, such as diving, gymnastics, ski jumping, cheerleading, etc. [4,8–10] or athletes who must maintain their balance while riding a horse over jumps and around obstacles [11,12], postural regulation is a key component of peak performance.

Although similar research has been conducted on female athletes [3], currently, there is a paucity of similar comparisons among male athletes. Due to the identified neuromuscular differences between men and women, such as deviations in proprioception, muscular activation patterns, muscular strength, and balance [13], there may similarly be differences in the postural control of male athletes between sports. Therefore, this study investigated the postural regulation and stability in male athletes from a variety of sports. More specifically, we examined the postural control between elite-level male athletes in diving, ice hockey, and soccer.

## 2. Materials and Methods

### 2.1. Subjects

Fifty male subjects volunteered to participate in this study. These professional athletes played in either the 3rd German league (ice hockey, soccer) or on an international level (diving). Informed consent was provided by all subjects before any testing, as approved by the institutional review board (reference number: 2016-70).

### 2.2. Study Design and Methodology

A posturographic measurement system (SensoDiaTrain (SDT), Haynl Elektronik, Schönebeck, Germany) was used to measure postural control. This device is based on the interactive balance system (IBS, neurodata GmbH, Vienna, Austria) (Figure 1a,b) was used to assess postural control. This system uses four force plates and samples at a rate of 32 Hz.

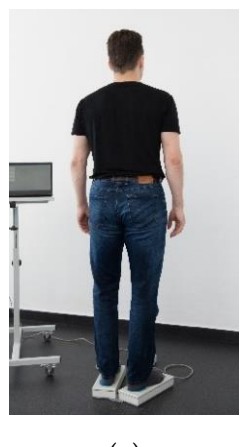 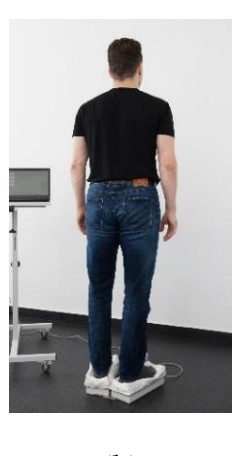

(**a**)  (**b**)

**Figure 1.** (**a**) stable force plates, (**b**) unstable force plates using foam pads.

Postural regulation was assessed during four balance tasks (Figure 1a,b). Each test scenario lasted 32 s, and participants were instructed to stand on both feet and remain as motionless, but not rigidly, as possible.

The four balance task conditions included: (1) eyes open, looking forward at a fixed target, and without foam pads (NO = Normally stand with Open eyes), (2) eyes closed, and without foam pads (NC = Normally stand with Closed eyes), (3) eyes open, looking forward at a fixed target, and with foam pads (PO = normally stand on foam Pads with Open eyes), and (4) eyes closed, and with foam pads (PC = normally stand on foam Pads with Closed eyes) [14–17]. The NO and NC tasks replicate the standard clinical assessment known as the Romberg test. The differences in tasks (eyes open/closed, stable/unstable base) were chosen to target different sensorimotor subsystems.

*2.3. Statistics*

All variables were first tested for the assumption of homogeneity of variance using the Levene test for equality of variances and the Shapiro—Wilk Test to test for normal distribution.

Between sport differences were determined using a one-factorial univariate general linear model. Covariates of age, height, and mass were used to determine the "pure" discrepancies in postural control and regulation. Bonferroni tests were then used for pairwise analyses. Dependent variables included stability indicator (ST), weight distribution index (WDI) and weight distribution on left extremity (left%), and weight distribution on heel (heel%). An alpha level of <0.05 was used for between-group statistical significance.

Partial eta squared ($\eta_p^2$) [18] was calculated to determine the effect sizes of the main effects ($\eta_p^2$) in order to estimate clinical significance and quantify differences between sports in postural control. Effect sizes were considered small ($\geq 0.01$), medium ($\geq 0.06$), or large ($\geq 0.14$) [19].

To assess the relationships between anthropometric and demographic characteristics with postural control, Pearson's product moment correlations were calculated. These relationships consisted as: <0.1 = trivial; 0.1–0.3 = small; 0.3–05 = moderate; 0.5–0.7 = large; 0.7–0.9 = very large; and 0.9–1.0 = nearly perfect [19].

SPSS version 28.0 for Windows (IBM, Armonk, NY, USA) was used to conduct all statistical analyses.

## 3. Results

*3.1. Normal Distribution and Homogeneity of Variance*

Abnormal distributions were found for ST_NO ($p = 0.008$), WDI_NO ($p = 0.009$), WDI_NC ($p = 0.025$), WDI_PC ($p = 0.002$) and all medio-lateral weight distribution parameters (LEFT_NO: $p < 0.001$; NC: $p = 0.001$; PO: $p = 0.014$; PC: $p = 0.035$).

Variance inhomogeneity (significant differences) were found for medio-lateral load distribution LEFT_NO ($p = 0.031$), LEFT_NC ($p = 0.039$), and anterior–posterior load distribution HEEL_NC ($p = 0.024$).

*3.2. Anthropometrics and Demographics*

The biggest significant differences ($p = 0.003$; Table 1) were noted for age among the divers (21.9 ± 4.46 years) and ice hockey players (27.6 ± 4.77 years). No significant differences were found between the ice hockey and soccer players ($p = 0.062$) or between the divers and soccer players ($p = 0.323$).

**Table 1.** Anthropometric subject data (mean ± standard deviation (SD)) by sport. Significant differences between sports ($\eta_p^2 > 0.10$) are indicated in bold.

| Sport | Age (years) | | Weight (kg) | | Height (m) | |
|---|---|---|---|---|---|---|
| | Mean | SD | Mean | SD | Mean | SD |
| Ice hockey ($n = 16$) | 27.6 | 4.77 | 85.4 | 6.43 | 1.81 | 0.05 |
| Soccer ($n = 23$) | 24.4 | 3.41 | 79.2 | 6.66 | 1.82 | 0.06 |
| Diving ($n = 11$) | 21.9 | 4.46 | 75.7 | 10.5 | 1.78 | 0.06 |
| $p/\eta_p^2$ | 0.003/0.217 | | 0.005/0.201 | | 0.175/0.071 | |

Divers had the lowest body weight (75.7 ± 10.5 kg), whereas ice hockey players displayed the highest body weight (85.4 ± 6.43 kg; $p = 0.006$). There was also a significant difference between ice hockey and soccer players ($p = 0.046$).

Conversely, body height did not show any between-group differences. The body height ranged from 1.78 m (divers) to 1.82 m (soccer players).

*3.3. Postural Control*

Postural control differed between sports (Table 2). The highest number (6) of performance maxima was found among the divers who also had the best postural stability

during tests with decreased somatosensory conditions (PO, PC). However, soccer players had the best stability during tests with standard or "normal" conditions (NO, NC) (Table 2; Figure 2).

**Table 2.** Descriptive comparison (mean ± standard deviation (SD)) between sports depending on test positions. Performance maxima per position and parameter marked in bold.

| Sports/Test Position | | ST | WDI | Left (%) | Heel (%) |
|---|---|---|---|---|---|
| Ice hockey (*n* = 16) | NO | 15.5 ± 3.51 | 4.56 ± 2.29 | **49.5 ± 2.23** | 53.8 ± 8.33 |
| | NC | 21.7 ± 5.56 | 4.64 ± 1.94 | **49.5 ± 1.92** | 52.1 ± 9.40 |
| | PO | 34.3 ± 4.92 | **2.98 ± 1.57** | **50.6 ± 3.01** | **48.4 ± 5.74** |
| | PC | 64.7 ± 10.3 | 2.94 ± 1.48 | 49.5 ± 3.02 | **48.5 ± 5.24** |
| Soccer (*n* = 23) | NO | **14.7 ± 5.04** | 4.20 ± 2.04 | 49.2 ± 2.90 | 48.2 ± 7.51 |
| | NC | **17.3 ± 5.66** | **4.07 ± 1.91** | 49.4 ± 2.80 | 46.4 ± 6.46 |
| | PO | 30.6 ± 7.21 | 4.27 ± 2.12 | 49.4 ± 3.11 | 43.5 ± 4.71 |
| | PC | 55.7 ± 12.1 | 4.16 ± 2.19 | **49.7 ± 3.46** | 43.5 ± 4.40 |
| Diving (*n* = 11) | NO | 16.6 ± 4.72 | **4.05 ± 2.65** | 52.6 ± 5.07 | **49.6 ± 6.86** |
| | NC | 24.9 ± 6.98 | 4.50 ± 2.46 | 53.1 ± 4.58 | **49.2 ± 7.45** |
| | PO | **26.2 ± 5.46** | 3.18 ± 2.64 | 51.8 ± 3.22 | 45.1 ± 5.08 |
| | PC | **54.6 ± 12.1** | **2.82 ± 2.13** | 52.3 ± 3.31 | 46.6 ± 4.38 |

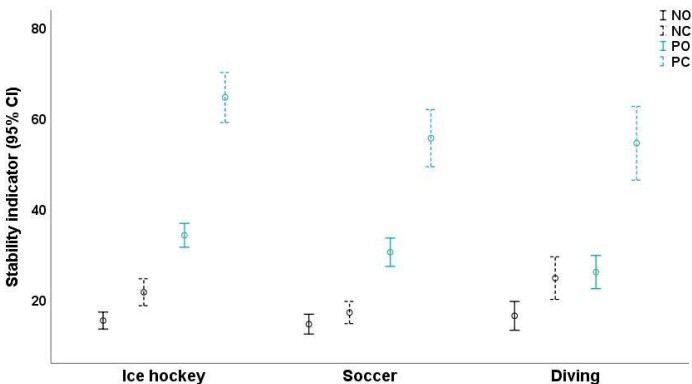

**Figure 2.** Stability indicators (ST) between sports during four test conditions (NO, NC, PO, PC). Values are provided as mean and 95% CI.

Ice hockey players demonstrated the highest level of medio-lateral weight distribution in three of four test positions (Table 2; Figure 3).

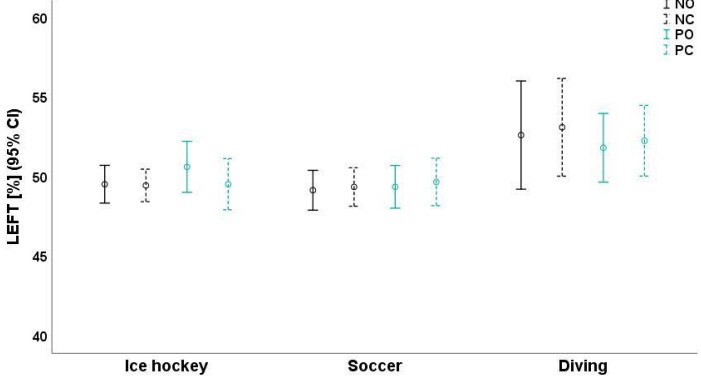

**Figure 3.** Medio-lateral load distribution (LEFT%) between sports during four test conditions (NO, NC, PO, PC). Values are provided as mean and 95% CI.

The anterior–posterior weight distribution (Table 2; Figure 4) was highest for diving athletes (NO, NC) and ice hockey players (PO, PC).

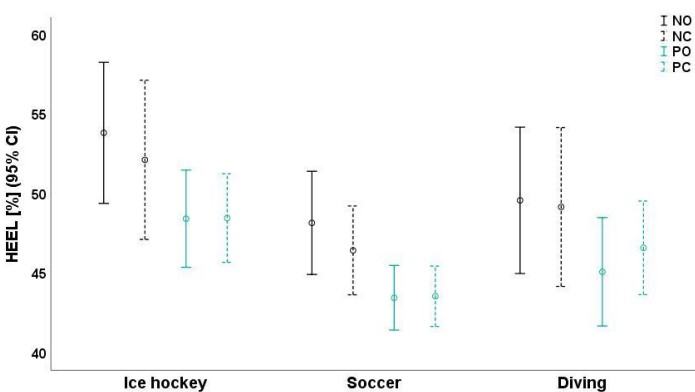

**Figure 4.** Anterior–posterior load distribution (HEEL%) between sports during four test conditions (NO, NC, PO, PC). Values are provided as mean and 95% CI.

The weight distribution index had the lowest differences between sports (Tables 2 and 3; Figure 5). In contrast to ice hockey players, the divers consistently showed the highest standard deviations during all test conditions (Figure 5; Table 2).

The ice hockey players showed the highest level of medio-lateral weight distribution (Table 2). Regarding weight distribution in the anterior–posterior direction, ice hockey players were the best subjects on pads (somatosensory input reduced) and diving athletes for the normal conditions (NO, NC; Table 2).

**Table 3.** Analysis of variance (covariates: age, mass, and height) between groups by test condition (eyes open, stable surface (NO, eyes closed, stable surface (NC, eyes open, unstable surface (PO, and eyes closed, unstable surface (PC) and postural control parameters. Significant differences ($\eta_p^2 \geq 0.14$) between sports indicated in bold.

| Variable/Test Position | | Effects/Variance Analysis ($p/\eta_p^2$) | | | |
|---|---|---|---|---|---|
| | | Sports | Covariates | | |
| | | | Age | Height | Weight |
| **ST** | NO | 0.413/0.04 | 0.411/0.02 | 0.282/0.03 | 0.675/0.01 |
| | NC | **0.001/0.27** | 0.890/0.00 | 0.093/0.06 | 0.785/0.01 |
| | PO | 0.085/0.11 | 0.926/0.00 | 0.720/0.01 | 0.381/0.02 |
| | PC | 0.062/0.12 | 0.736/0.01 | **0.008/0.15** | 0.554/0.01 |
| **WDI** | NO | 0.647/0.02 | 0.020/0.12 | 0.330/0.02 | 0.092/0.06 |
| | NC | 0.340/0.05 | 0.030/0.10 | 0.874/0.01 | 0.861/0.01 |
| | PO | 0.222/0.07 | 0.597/0.01 | 0.962/0.00 | 0.762/0.01 |
| | PC | 0.177/0.08 | 0.752/0.02 | 0.133/0.05 | 0.018/0.12 |
| **HEEL** | NO | 0.129/0.09 | 0.813/0.01 | 0.308/0.02 | 0.848/0.01 |
| | NC | 0.206/0.07 | 0.801/0.01 | 0.626/0.01 | 0.291/0.03 |
| | PO | **0.031/0.15** | 0.257/0.03 | 0.834/0.01 | 0.683/0.00 |
| | PC | 0.094/0.10 | 0.272/0.03 | 0.075/0.07 | 0.031/0.10 |
| **LEFT** | NO | 0.133/0.09 | 0.140/0.05 | 0.363/0.02 | 0.556/0.01 |
| | NC | **0.021/0.16** | 0.291/0.03 | 0.978/0.00 | 0.709/0.01 |
| | PO | 0.098/0.01 | 0.639/0.01 | 0.494/0.01 | 0.863/0.01 |
| | PC | 0.284/0.06 | 0.080/0.07 | 0.732/0.01 | 0.494/0.01 |

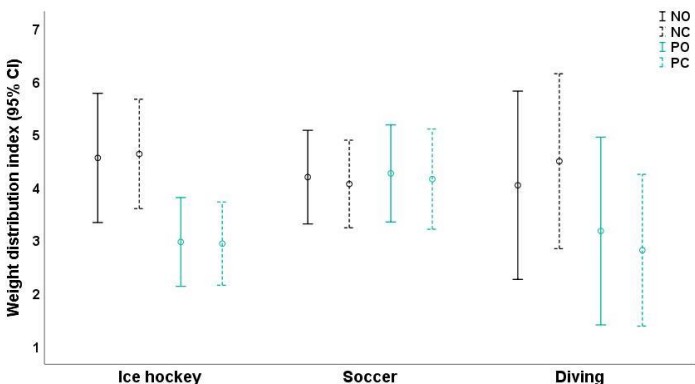

**Figure 5.** Weight distribution index between sports during four test conditions (NO, NC, PO, PC). Values are provided as mean and 95% CI.

A total of 75% (3/4) of the significant differences (predictors: sports, age, height, and weight) were provided by sport (Table 3). ST_NC had the largest effect size ($p = 0.001$, $\eta_p^2 = 0.273$), while the stability indicator (ST) had the most significant effects (50%, 2/4). In contrast, medio-lateral load distribution (LEFT) showed only one significant effect between groups during test position NC (Table 3).

Significant differences between sports partial effects were found for:

- ST-NC: soccer vs. diving: $p = 0.001$,
- HEEL-PO: ice hockey vs. soccer: $p = 0.027$,
- LEFT-NC: soccer vs. diving: $p = 0.019$.

### 3.4. Relationships between Anthropometrics and Demographics with Postural Control

The relationships between anthropometrics and demographics with postural control can be viewed in Table 4. There were no relevant ($r > 0.5$) correlations noted. The correlation coefficients ($r$) ranged from $-0.01$ (PO_HEEL_Age) to 0.43 (PC_ST_Height) (Table 4).

**Table 4.** Bivariate correlations ($r$) between anthropometrics and demographics with posture control parameters (ST, WDI, HEEL, LEFT) under four testing conditions (NO, NC, PO, PC).

| Test Position | Parameters | Age | Height | Weight |
|---|---|---|---|---|
| **NO** | ST | −0.20 | 0.15 | 0.02 |
| | WDI | −0.21 | 0.11 | 0.19 |
| | HEEL | 0.10 | 0.16 | 0.21 |
| | LEFT | −0.27 | −0.16 | −0.13 |
| **NC** | ST | −0.09 | 0.15 | 0.12 |
| | WDI | −0.29 | 0.11 | 0.04 |
| | HEEL | 0.13 | 0.18 | 0.34 |
| | LEFT | −0.29 | −0.05 | −0.10 |
| **PO** | ST | 0.25 | 0.11 | 0.30 |
| | WDI | −0.12 | 0.02 | −0.12 |
| | HEEL | −0.01 | 0.03 | 0.17 |
| | LEFT | −0.13 | 0.05 | 0.02 |
| **PC** | ST | 0.07 | 0.43 | 0.34 |
| | WDI | −0.17 | 0.07 | −0.28 |
| | HEEL | 0.08 | −0.09 | 0.26 |
| | LEFT | −0.39 | −0.04 | −0.22 |

## 4. Discussion

Because different sports require different kinematics and kinetics to be competitive, we hypothesized that there would be differences in postural control between sports as well. More specifically, divers would present with better postural regulation compared to

hockey and soccer players. This hypothesis was supported, as our results indicated that both ice hockey and soccer players had poorer posturographic performance compared to divers. This was especially prevalent during the more demanding postural positions that attempted to limit the somatosensory system by using an unstable surface and during the eyes closed tasks (PO, PC).

Soccer players had their best postural stability under "normal conditions" (NO, NC). In return, the divers performed best under reduced somatosensory conditions (PO, PC). This result is not surprising since the sport-specific movements of divers and their subsequent postural control are much more demanding than soccer and hockey players, who are typically not required to produce aerial rotations. Our findings support previous research, which has suggested that more demanding test positions demonstrate greater between-group differences [20]. These results also partially support the findings of Lauenroth et al. [3], who assessed the postural stability of female athletes from a variety of sports and found that gymnasts had superior control over the athletes from the team-based sports. Davlin [21] used a stabilometer to assess dynamic balance and found that gymnasts had better postural control than soccer players and swimmers. Our results suggest that male athletes have similar postural control differences between sports.

Schwesig et al. [22] investigated 1724 asymptomatic individuals over the life span (6–97 years). The gender stratified findings of the multifactorial regression analysis revealed that age, mass, and height have a non-linear relationship with postural control. Similarly, other research has suggested that when using stabilometric parameters to assess postural control, clinicians should include anthropometric characteristics [23]. Previous research has also reported that postural control related to physical characteristics is different between sexes [24]. For example, taller and heavier women present with less postural stability. Conversely, taller and heavier men actually present with better postural control. As such, some research has suggested considering subsequent factors when testing, such as activity level [25], which is why our study only tested elite-level athletes.

The weight distribution index had the smallest differences between sports. Ice hockey players showed the highest medio-lateral weight distribution in three out of four test positions. The divers presented with the best stability indicator during the more difficult test conditions. These findings support previous research that has shown similar postural control among both male [21] and female [2,21] technical athletes when compared to team sport athletes. In female athletes, sports that require technical kinematics, such as diving and ski jumping, were found to have significantly better postural stability compared to athletes from team sports. However, this does not suggest that postural control should not be considered in the physical training of athletes in team sports. Previous research has repeatedly reported that deficient postural regulation may lead to injury, especially in team sport athletes that require cutting maneuvers and quick accelerations and decelerations [5,13]. Fortunately, Andreeva et al. [26] found that an increased volume of training, regardless of sport, subsequently increased postural control.

*Limitations*

As with any research, there are a few limitations of our study worth noting. First, we tested postural control using a static activity. However, an important physical characteristic of divers, soccer players, and hockey players is strong dynamic performance. This may have resulted in test conditions that did not mimic the athlete's respective sport-specific conditions. Our testing protocol was chosen due to its' clinical relevance and to accurately compare previous findings with our results that used a similar methodology. Second, differences in training routines between the athlete groups may have had an effect on the results of our study. Third, and as previously mentioned, the amount of physical training in an individual's respective sport can directly impact their level of postural control. Optimal training programs may be easier for coaches to create when working with individual sports such as diving, compared to team sports such as soccer and hockey, which may also be influenced by the physical requirements of each field-specific position. In team sports, core

stability and mobility training are often less frequently used when compared to technical sports. This can lead to imbalances in key muscle groups and an increased risk of injury. Lastly, our groups of subjects had significantly different sample sizes (ice hockey: $n = 16$, soccer: $n = 23$, diving: $n = 11$), which may result in an overestimation of differences; [27] thereby limiting the interpretation of results. In this context, we did not perform a power analysis to calculate the number of necessary subjects because the number of athletes was limited based on the performance level and the voluntary nature of the study. Therefore, it was a sample of convenience.

### 5. Conclusions

Divers showed better postural control compared to soccer and ice hockey players. These differences were predominantly found during the more challenging balance conditions. Under non-perturbated conditions, the soccer players showed the highest postural stability. Unfortunately, these easier conditions are not typically present during soccer practice and games.

Based on these results, balance training should be incorporated into athletes participating in team sports, such as soccer and ice hockey, in an attempt to improve performance and decrease the risk of injury. Periodic testing of postural control should also be considered to measure potential improvements or developed deficiencies among athletes. Our findings can help athletes and coaches create and adjust training regimens to aid in increased performance and the decreased risk of injury.

*Practical Implications*

Coaches and athletes often perform subjective assessments of players' athletic performance, which can be very informational. However, this type of analysis also has its' limits. As such, providing players with objective feedback can further expand their athletic performance capabilities. The results of our study suggest that male athletes from soccer, ice hockey, and diving have significant differences in postural control as measured using a clinically assessable posturographic device. Whether this type of instrumentation is available to respective coaches and athletes or not, our results provide valuable information to help increase sports performance and diminish the risk of injury. Coaches should consider specific training for postural control, especially in more technical sports such as diving. However, the relationship between postural control among these athletes and the direct effect on sports performance and potential injury needs further investigation.

**Author Contributions:** Conceptualization, A.L., L.R. and R.S.; methodology, A.L., L.R., K.G.L. and R.S.; formal analysis, A.L., L.R., S.S. and R.S.; investigation, A.L., L.R., S.S. and R.S.; resources, A.L.; data curation, A.L. and L.R.; writing—original draft preparation, A.L., L.R. and R.S.; writing—review and editing, A.L., L.R., S.S. and R.S.; visualization, L.R. and S.S.; supervision, K.-S.D. and R.S.; project administration, A.L. and R.S. All authors have read and agreed to the published version of the manuscript.

**Funding:** This study was supported by the federal state of Saxony-Anhalt (number: 1604/00017) and the European Regional Development Fund (ERDF 2014-2020). We acknowledge the financial support within the funding program for Open Access Publishing by the German Research Foundation (DFG).

**Institutional Review Board Statement:** The study was conducted according to the guidelines of the Declaration of Helsinki and approved by the local Ethics Committee of the Martin-Luther-University Halle-Wittenberg (reference number: 2016-70; date of approval: 12 October 2016). All procedures performed in studies involving human participants or on human tissue were in accordance with the ethical standards of the institutional and/or national research committee and with the 1975 Helsinki declaration and its later amendments or comparable ethical standards.

**Informed Consent Statement:** Informed consent was provided by all study participants.

**Data Availability Statement:** The raw data supporting the conclusions of this article will be made available by the authors, without undue reservation.

**Acknowledgments:** The authors would like to thank the athletes and coaches for their participation in our study.

**Conflicts of Interest:** The authors declare no conflict of interest.

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
