# Peer review of "Comparison of Postural Stability and Regulation among Male Athletes from Different Sports"

_applsci, doi:10.3390/app12115457_

Round 1

Reviewer 1 Report

You don`t have enough quotes in the introduction

I don`t think you`ve discovered these postural differences between disciplines

You need to have more opinions from other authors who have done research on this topic

The results of the research between diving and hockey have been incomparable to football because it is not a sport that has daily training. 

Finally, the article can be published without changing the research part, only the introductory part needs to be improved.

The conclusions are consistent with the evidence and arguments presented.   

Author Response

Response to Reviewer 1 Comments

(red=comment to the editor/reviewer; blue=changes in the manuscript)

INTRODUCTION

Point 1: You don`t have enough quotes in the introduction. You need to have more opinions from other authors who have done research on this topic. Finally, the article can be published without changing the research part, only the introductory part needs to be improved.

Response 1:

Thank you for this feedback. Within the last rework, we checked the literature and publications (Medline, PubMed, Google Scholar), but we did not find any new suitable reference.

RESULTS

Point 2: I don`t think you`ve discovered these postural differences between disciplines.

Response 2:

Thank you for this valuable comment. According to line 94, we adapted/ enhanced the relevance criteria (ηp2≥0.14, see legend of table 3, line 198) to improve the readability and clarity of the results. Now, it should be clearer, that we only found few differences between sports (soccer vs. diving and soccer vs. ice hockey), because the number of main effects is equal compared with the partial effects.

       @ line 189-191: 75% (3/4) of the significant differences (predictors: sports, age, height, and weight) were provided by sport (Table 3). ST_NC had the largest effect size (p = 0.001, ηp2 = 0.273), while the stability indicator (ST) had the most significant effects (50%, 2/4).

DISCUSSION

Point 3: The results of the research between diving and hockey have been incomparable to football because it is not a sport that has daily training.

Response 3:

We chose these sports because the very different requirements regarding posture stability and regulation. Of course, the content of training and workout is completely different but the aspect of daily training is equal. We added for a better understanding the following paragraph:

       @ line 182-185: The ice hockey players showed the highest level of mediolateral weight distribution (Table 2). Regarding weight distribution in the anterior-posterior direction, ice hockey players were the best subjects on pads (somatosensory input reduced) and diving athletes for the normal conditions (NO, NC; Table 2).

CONCLUSION

Point 4: The conclusions are consistent with the evidence and arguments presented.

Response 4:

Thanks a lot for this positive feedback.  

Reviewer 2 Report

Firstly, I would like to recognize the authors for evaluating postural regulation and stability in male athletes of different sports using posturography.

The title is clear, concise, and informative.

The abstract is clear and includes the objectives, design, methods, variables considered, main results and most relevant conclusion. Minor issues such as “one individual sports” can be easily modified after proofing the manuscript.

Line 21: I would say sport instead of sports.

Line 24: Performance maxima. I am not sure what you mean here.

The introduction is well presented and creates a good rationale for the purpose of the study. 

Methods section

Line 70: I would say participants instead of test persons.

 Lines 71-72: “played in the 3rd German league (ice hockey, soccer) and moved on international level 71 (diving)”. I am not sure what you mean here.

Line 77: “the female athletes were tested using “. In the abstract, you stated that you have male subjects, while you are referring to female participants here.

Did you do a power analysis to identify the number of subjects, or was this a convenience sample?

Results section

Lines 126-132: You use abbreviations, but you have not presented those earlier. First, spell them out and then use the abbreviations so that the reader can follow.

Line 223: What do you mean by dependences ?

 Discussion section

 The discussion section presents and analyses the results accurately—however, some paragraphs consist of only two sentences. I would improve the structure and flow of the discussion section. The limitations are presented as well as the practical applications.

Author Response

Response to Reviewer 2 Comments

(red=comment to the editor/reviewer; blue=changes in the manuscript)

COMMON and TITLE

Point 1: Firstly, I would like to recognize the authors for evaluating postural regulation and stability in male athletes of different sports using posturography. The title is clear, concise, and informative.

Response 1:

Thank you for this positive feedback.

ABSTRACT

Point 2: The abstract is clear and includes the objectives, design, methods, variables considered, main results and most relevant conclusion. Minor issues such as “one individual sports” can be easily modified after proofing the manuscript.

Line 21: I would say sport instead of sports.

Response 2:

Corrected as suggested at first (sports instead of sport):

            @ line 21: … and one individual sports …

Point 3: Line 24: Performance maxima. I am not sure what you mean here.

Response 3:

Performance maxima means the highest level of performance concerning several parameters.

INTRODUCTION

Point 4: The introduction is well presented and creates a good rationale for the purpose of the study. 

Response 4:

Thank you for this positive feedback and appreciation of our work in this part of the manuscript.

METHODS

Point 5: Line 70: I would say participants instead of test persons.

Response 5:

Corrected as suggested:

            @ line 68-69: For each test scenario lasted 32 seconds and participants were instructed …

Point 6: Lines 59-60: “played in the 3rd German league (ice hockey, soccer) and moved on international level (diving)”. I am not sure what you mean here.

Response 6:

The diving athletes performed on an international level. For this kind of sport, we cannot distinguish different leagues comparable with team sports (league 1, 2 and 3). Therefore, we used the category national vs. international in order to emphasize the high and professional performance level of the diving athletes. For more clarification, we changed the sentence as follows:

            @ line 59-60: These professional athletes played in either the 3rd German league (ice hockey, soccer) or on an international level (diving).

Point 7: Line 77: “the female athletes were tested using “. In the abstract, you stated that you have male subjects, while you are referring to female participants here.

Response 7:

Thanks a lot for this valuable hint. We corrected this mistake as follows:

            @ line 59: 50 male subjects participated in this study. These professional athletes played in …

Point 8: Did you do a power analysis to identify the number of subjects, or was this a convenience sample?

Response 8:

We did not perform a power analysis to calculate the number of necessary subjects, because the number of athletes is limited based on the performance level and the voluntary character of the study. Therefore, it was a convenience sample. We added this information in the limitations of this manuscript as follows.

            @ line 272-275: In this context, we did not perform a power analysis to calculate the number of necessary subjects, because the number of athletes is limited based on the performance level and the voluntary character of the study. Therefore, it was a convenience sample.

RESULTS

Point 9: Lines 126-132: You use abbreviations, but you have not presented those earlier. First, spell them out and then use the abbreviations so that the reader can follow.

Response 9:

Corrected at the first mention as suggested:

            @ line 72-77: The four balance task conditions included: 1) eyes open, looking forward at a fixed target, and without foam pads (NO = Normally stand with Open eyes), 2) eyes closed, and without foam pads (NC = Normally stand with Closed eyes), 3) eyes open, looking forward at a fixed target, and with foam pads (PO = normally stand on foam Pads with Open eyes), and 4) eyes closed, and with foam pads (PC = normally stand on foam Pads with Closed eyes) [14-17].

Point 10: Line 223: What do you mean by dependences?

Response 10:

I think, this is a duplication that we removed.

            @ line 204-205: The relationships between anthropometrics and demographics with postural control can be viewed in Table 4. There were no relevant (r>0.5) correlations noted.

DISCUSSION

Point 11: The discussion section presents and analyses the results accurately—however, some paragraphs consist of only two sentences. I would improve the structure and flow of the discussion section. The limitations are presented as well as the practical applications.

Response 11:

We improved the discussion including limitations. Now, all paragraphs are longer than two sentences and with a consistent structure.
